# Techniques and Outcomes of Endoscopic Ultrasound Guided—Pancreatic Duct Drainage (EUS- PDD)

**DOI:** 10.3390/jcm12041626

**Published:** 2023-02-17

**Authors:** Jun Liang Teh, Anthony Yuen Bun Teoh

**Affiliations:** 1Department of Surgery, Juronghealth Campus, National University Health System, Singapore 609606, Singapore; 2Department of Surgery, Prince of Wales Hospital, The Chinese University of Hong Kong, Hong Kong

**Keywords:** EUS-guided pancreatic duct drainage, surgically altered anatomy, transmural drainage, EUS-rendezvous ERCP

## Abstract

Endoscopic ultrasound guided—pancreatic duct drainage (EUS- PDD) is one of the most technically challenging procedures for the interventional endoscopist. The most common indications for EUS- PDD are patients with main pancreatic duct obstruction who have failed conventional endoscopic retrograde pancreatography (ERP) drainage or those with surgically altered anatomy. EUS- PDD can be performed via two approaches: the EUS-rendezvous (EUS- RV) or the EUS-transmural drainage (TMD) techniques. The purpose of this review is to provide an updated review of the techniques and equipment available for EUS- PDD and the outcomes of EUS- PDD reported in the literature. Recent developments and future directions surrounding the procedure will also be discussed.

## 1. Introduction

Endoscopic ultrasound (EUS) pancreatic duct drainage (EUS- PDD) was first described by Bataille et al. [1] in 2002 when they reported EUS-guided main pancreatic duct (MPD) puncture for trans-duodenal rendezvous endoscopic retrograde pancreatography (ERP). Francois et al. went on to report a series of four patients who underwent EUS-guided pancreaticogastrostomy (EUS- PG) of which three out of four patients achieved satisfactory relief of pain at 1-year follow-up [2]. Currently, EUS- PDD remains a challenging yet infrequently performed procedure for the advanced interventional endoscopist. EUS- PDD serves as a rescue procedure to access the pancreatic duct when standard transpapillary ERP has failed or is not possible due to altered anatomy. Retrospective series have demonstrated that the frequency of EUS- PDD is uncommon even in specialized tertiary referral centers for pancreatic diseases, ranging between two and four cases per center per year [3,4,5]. Through this review, we aim to provide the reader with an update on the indications, accessories required, and techniques employed as well as a summary of the literature surrounding EUS- PDD outcomes.

## 2. Indications for EUS- PDD

EUS- PDD is indicated for patients with symptomatic obstruction of the MPD that is not amenable to conventional ERP drainage. The main indications and contraindications for EUS- PDD are summarized in Table 1. MPD obstruction may occur because of multiple etiologies, including fibrosis and inflammation in chronic pancreatitis, obstructing pancreatic duct stones, or malignant obstruction. Pancreatic outflow obstruction resulting in ductal hypertension is more commonly seen in benign pancreatic obstruction, and less in infiltrative malignant disease, where the need for pancreatic duct decompression is rare. EUS- PDD is also helpful for pancreatic duct drainage for pancreatico-jejunostomy anastomotic strictures (PJAS) after pancreatoduodenectomy. EUS- PDD is also indicated for the management of symptomatic pancreatic duct (PD) stones and disconnected PD syndrome (DPDS). Recently, the use of EUS- PDD to manage intractable post-operative pancreatic fistula has been described [6]. EUS- PDD is a valuable technique to gain access to the pancreatic duct when the endoscopist is unable to cannulate the PD or when the PD is inaccessible due to altered post-surgical anatomy [7]. In a recent review of 2205 cases of pancreaticobiliary ductal access and drainage, Garcia- Alonso et al. reported that 107 endoscopic procedures were performed for pancreatic indications. A total of 10% eventually (n = 11) required EUS- PDD (eight transmural stenting and three EUS- RV) [8]. Four of these procedures were undertaken directly either due to anticipated failure or altered surgical anatomy and seven were undertaken after failed ERCP or performed as a combination procedure with ERCP [8]. In the management of DPDS, EUS-guided PDD aims to drain the viable upstream pancreas with a plastic stent. When direct puncture of the pancreatic duct is difficult, Ghandour et al. described two cases where the patients underwent a modified approach, with EUS-guided drainage of the fluid collection in communication with the disrupted MPD, resulting in successful symptom resolution with no recurrence of acute pancreatitis on short term follow up [9]. Although rarely indicated, EUS-guided PDD has also been described following obstructive pancreatitis in a patient with ampullary adenocarcinoma where transpapillary PD drainage was not successful [10].

## 3. Contraindications for EUS- PDD

Contraindications for EUS- PDD include patient factors such as hemodynamic instability and bleeding risk due to severe coagulopathy or severe thrombocytopenia. Technical contraindications include the inability to localize the MPD on EUS or when the PD is insufficiently dilated (<4 mm), the presence of intervening vessels prohibiting safe puncture of the MPD, and the presence of multi-level strictures. There are no published data on the minimum size of PD dilatation for a successful puncture, but most series quote a median PD diameter between 4 and 6 mm [11,12,13]. Technical failure for PD puncture has been reported in cases of non-dilated PD measuring 2 mm [11]. In a meta-analysis consisting of 22 studies involving 714 patients, the mean MPD diameter was between 3.5 and 8.1 mm [14]. Based on these results, main PD dilatation of at least 4 mm will likely increase the chance of successful PD puncture. EUS- PDD is also contraindicated where standard ERP techniques to access the PD have not been exhausted as EUS- PDD is more invasive and may be associated with greater risk of adverse outcomes compared to standard ERP techniques [8,15]. 

It should be emphasized EUS- PDD is a challenging procedure and should only be attempted by interventional endoscopists skilled in both ERCP and EUS. EUS- PDD is challenging for the following reasons [3]:a.difficult puncture with standard needles due to the small caliber of the dilated PD embedded within a fibrotic pancreas;b.unstable scope position and consequent poor force transmission to the puncturing needle;c.difficult wire manipulation through the needle (due to PD stricture, unfavorable needle to duct angle, preferential passage into a dilated side branch [7]), and risk of wire shearing during manipulation;d.difficulty passing devices such as balloons or cystotomes into the PD for tract dilatation;e.fragility of the pancreas and associated adverse events after aggressive manipulation.

In a Spanish national survey of 19 centers with limited experience in EUS-guided cholangiopancreatography, technical success rates of pancreatic EUS-guided cholangiopancreatography (ESCP) was lower than biliary ESCP (57.9% vs. 68.9%) with higher complication rates (26.3% vs. 22.6%) [15]. The main reason for technical failure was failed manipulation of the guidewire inside the pancreatic or biliary duct [15].

## 4. Technique of EUS- PDD

### 4.1. Patient Preparation

We perform EUS- PDD with the patient in a prone position under general anesthesia or monitored anesthetic care [7]. Positioning the patient in the supine position for EUS- PDD has also been described [16]. Prophylactic antibiotics are administered [7,16] prior to the start of the procedure. Anticoagulation should be corrected, and anti-thrombotic medications withheld as outlined by published guidelines [17].

### 4.2. Approaches and Equipment

EUS- PDD comprises two approaches to drain the pancreatic duct [13] including EUS-assisted rendezvous (EUS- RV) ERP and EUS-transmural drainage (EUS- TMD) [18]. In scenarios where the papilla is accessible with a standard therapeutic duodenoscope but cannulation of the pancreatic duct by ERP has failed previously, EUS- RV is the procedure of choice due to higher success rates, lower adverse events, and avoidance of an extra-anatomical stent [16,19,20]. Stenting performed by EUS- TMD can be either antegrade or retrograde or transpapillary/trans-anastomotic [18]. The puncture site for the pancreatic duct is usually from the stomach (EUS-pancreaticogastrostomy, EUS- PGS) or the duodenum (EUS-pancreaticoduodenostomy, EUS- PDS). Typically, to access the pancreatic duct at the body and tail, EUS- PGS is more favorable whereas EUS- PDS is more suitable for drainage of the pancreatic head. The choice of technique will depend on the accessibility of the native papilla, the ability to pass the guidewire into the PD and maneuver it across the stricture as well as the desired direction of stent drainage. These considerations are summarized in Figure 1.

### 4.3. Pancreatic Duct Access

The steps common to all approaches of EUS- PDD are first pancreatic duct puncture under EUS guidance, followed by contrast injection and guidewire placement. To gain access to the pancreatic duct, a curvilinear array echoendoscope is inserted into either the stomach or duodenal bulb and EUS performed to identify the dilated pancreatic duct. Factors affecting the choice of puncture site will include the distance between the visceral lumen and PD, the presence of intervening vessels, and the stability of scope position which facilitates subsequent tract dilatation and stent deployment [7]. The most common puncture site for PD access is the stomach. It is recommended that the pancreatic duct is punctured at an oblique angle rather than perpendicularly as the latter will result in a difficult insertion of the guidewire and subsequent device insertion into the pancreatic duct [18].

Using a 19G needle (EZ Shot 3 plus, Olympus Medical Systems, Tokyo, Japan), the PD is punctured and a diagnostic pancreatogram is performed. A 0.035-inch or 0.025-inch (VisiGlide2; Olympus Medical Systems) guidewire is inserted into the pancreatic duct, directed either toward the head of the pancreas or the tail depending on the subsequent plan. In cases where the MPD is minimally dilated or the pancreatic parenchyma is fibrotic, a 22 G needle and a corresponding 0.018-inch or 0.021-inch guidewire can be used [21]. However, 0.018 in wires lack stability for over-the-wire exchange of accessories and these wires will need to be exchanged for larger caliber guidewires after initial manipulation. Guidewire shearing has been reported during guidewire manipulation in EUS- PDD; tips to avoid guidewire shearing will include gentle manipulation of the guidewire, avoiding withdrawal of guidewire back into the needle tip, as well as retraction of the needle tip into the sheath or the echoendoscope during guidewire manipulation [18]. Allowing excess length of the guidewire in the distal pancreatic duct or forming loops in the duodenum prevents loss of access to the PD. When wire manipulation into the duodenum is unsuccessful, injection of diluted methylene blue (1–3 cc of methylene blue, diluted with 15 cc saline or contrast) via the FNA needle into the duct may help to endoscopically identify an obscured ampullary orifice to aid cannulation by the duodenoscope during EUS- RV [22,23].

### 4.4. EUS- RV ERP

EUS- RV ERP is indicated when the papilla is still anatomically accessible by the duodenoscope, but initial cannulation of PD by conventional ERP has failed (Figure 1). Following pancreatic duct puncture, the guidewire is introduced antegrade towards the head of the pancreas into the duodenum and allowed to form loops in the duodenum. The linear echoendoscope is removed, leaving the guidewire in situ. The therapeutic duodenoscope is then inserted in the usual fashion into the second part of the duodenum and the end of the previously inserted guidewire is grasped into the working channel of the duodenoscope using forceps or a snare and withdrawn through the accessory channel. From here, the PD can either be cannulated over the wire or alongside the wire. In patients with pancreatic divisum, the guidewire may be introduced through the pancreatic duct and directed into the duodenum via the minor papilla [24]. This allows for papillotomy of the minor papilla and insertion of a 5 cm 10 Fr plastic stent for drainage of the pancreatic duct.

### 4.5. Transmural Approaches with Transpapillary or Trans-Anastomotic Stenting

Transmural techniques are employed for either EUS- PGS or EUS- PDS or transpapillary/trans-anastomotic stenting (Figure 2). Transmural EUS- PDD is performed when the papilla is inaccessible or when EUS RV is not successful l [21]. The MPD is accessed in a similar fashion as described above. The steps after initial access include dilatation of the EUS- PDS or PGS fistula track followed by deployment of the stent. Dilatation of the puncture site can be performed either with cautery or non-cautery devices [19]. Non-cautery dilatators will include either mechanical dilators or balloon dilators (Hurricane RX Biliary Dilatation Balloon, Boston Scientific, Malborough, MA, USA). Options for cautery dilators include the use of either a 6 Fr cystotome (Cysto Gastro Set; Endo-flex, GmbH, Voerde, Germany) or triple lumen needle knife (Microknife; Boston Scientific) [21]. Tract dilatation is a difficult step in EUS- PD; the endoscopist may encounter difficulty penetrating the fibrotic pancreatic parenchyma when a mechanical dilator or balloon dilator is employed for track dilatation without cautery. Conversely, several groups advocate the use of mechanical dilatators [25,26] before using cautery-assisted devices due to the risk of bleeding associated with cautery devices [27]. Recently, devices have become available. A new ultra-tapered mechanical dilator (ES dilator DC7R180S; Zeon Medical Co., Ltd., Tokyo, Japan) developed for EUS- PDD demonstrated decreased bleeding risk (0% vs. 18.2%, *p* = 0.04) with similar rates of dilatation success (93.3% vs. 95.0%, *p* < 0.05) when compared to a 6Fr cautery dilator [28]. Nakai et al. described a double-wire technique in 2019 to help stabilize the echoendoscope position when performing EUS-guided pancreatic stent placement for a patient with PJ stenosis [29]. With 0.025 in guidewire in place, fistula track dilatation with 6 Fr cystotome was performed. A double-lumen catheter (double-lumen cannula; Piolax, Kanagawa, Japan) was inserted, and an additional 0.035 in guidewire inserted (Renowave Ultrahard; Piolax) through the PJ stricture into the jejunum [29]. The introduction of the additional guidewire reduces the angulation between the puncture site and the PD which facilitates the insertion of the stent device and other accessories. Two plastic stents were then placed separately over the two guidewires. On the other hand, a plastic stent can also be deployed across the papilla or PJ anastomosis, with the distal end of the stent in the jejunum and the proximal end of the stent in the gastric lumen (i.e., “ring drainage” or gastro-pancreatico-jejunostomy) [5,30]. Ring drains facilitate future stent exchanges by keeping the pancreatico-gastric fistula track patent and reduces the risk of stent migration risk [30].

### 4.6. Transmural Stenting with Antegrade/Retrograde Stenting

Transmural antegrade or retrograde stenting is performed when there is difficulty passing the guidewire beyond the pancreatic duct stricture or PJ anastomotic stricture. Transmural stenting can be either antegrade when the stent is placed towards the head of the pancreas or retrograde when placed towards the tail of the pancreas. Following pancreatic duct access and fistula track dilatation as described above, plastic stents (usually 5 or 7 Fr plastic stents) are deployed with the distal end in the pancreatic duct and the proximal end in the enteric lumen. When choosing plastic stents for the creation of PDS or PGS, one should choose a plastic stent without a side aperture (Tannenbaum type) to avoid pancreatic juice leakage into the peritoneum via the side hole [31]. The use of modified anti-migratory FCSEMS with proximal and distal anchoring flaps (M.I. Tech, Seoul, Korea) across the pancreatico-gastric or enteric anastomosis has also been described [32], although there is a risk of obstructive pancreatitis if side branches of the PD are blocked. Success rates and long-term outcomes following FCSEMS in patients with pancreatico-jejunostomy anastomotic strictures (PJAS) following a Whipple operation have also been described [33]. Out of 23 patients who underwent FCSEMS, 5 patients (21.7%) developed late adverse events of which only 1 (4.3%) was due to stent occlusion which resulted in symptom recurrence [33]. Uncovered SEMS is not used due to the risk of pancreatic fluid leakage between the stomach and the pancreas.

To circumvent the problem of difficult fistula track dilatation and subsequent passage of stent across the gastric wall and pancreatic parenchyma into the MPD, Hayat et al. described PGS creation using small caliber accessories. After EUS-guided puncture of the PD and pancreatogram, over 0.018 inch guidewire, a 4 Fr angiogplasty balloon (Sterling, Boston Scientific, Malborough, MA, USA) was used to dilate the pancreatico-gastrostomy fistula track. A 3 Fr single pigtail stent was used to drain the pancreatic duct with the pigtail end in the intestine lumen or proximal PD, and the proximal end in the gastric lumen [34].

### 4.7. Per Oral Pancreaticoscopy

Per-oral pancreaticoscopy (POPS) following the creation of pancreatico-gastrostomy (PGS) and stent insertion has been described [35]. Three months following the PGS creation, a guidewire is inserted alongside the previously inserted stent and the stent is removed. The fistula and stricture sites were dilated with a 4 mm balloon (4-mm REN; Kaneka). A digital cholangiopancreatoscope (Spy Scope DS; Boston Scientific) was used to evaluate the cause of the stricture and perform electrohydraulic lithotripsy (EHL) of any pancreatic stones. The plastic stents were then regularly exchanged every two to three months for a year.

Per-oral pancreaticoscopy was performed in 13 out of 19 patients who underwent EUS- PGS [35]. Technical success of POPS was 100% with the median length of the procedure taking 66 min. POPS resulted in mild pancreatitis in one patient (8%), and asymptomatic stent migration in another. Following POPS, two or three 7-Fr plastic stents were placed across the stricture for dilatation. At the time of data analysis, four patients with benign fibrotic pancreatic strictures who had reached 1-year follow-up had improvement in pancreatic duct strictures and were stent free.

## 5. Outcomes

The outcomes of patient cohorts undergoing EUS- PDD are summarized in Table 2. Kahaleh reported in 2007 a series of 13 patients who underwent EUS-guided PGS with a plastic stent, of which 7 patients had surgically altered anatomy and the rest had pancreatic duct stricture due to pancreatitis or neoplastic process [4]. A total of 10 patients had successful stent placement across the PGS fistula (76.9%). After a mean follow-up of 14 months, mean pancreatic duct size was reduced from 4.6 to 3.0 mm (*p* = 0.01) with significant improvement in pain score from 7.3 to 3.6 (*p* = 0.01) [4]. Adverse events occurred in two patients, one with bleeding and another with contained perforation. Will et al. described a series of 12 patients over a 3-year period who underwent EUS- PGS after failing standard ERP and drainage of the pancreatic duct [36]. Pancreatography was successful in all patients and drainage was achieved in 69% of patients [36]. The adverse events rate was 42.9% with post-procedural pain accounting for most events. A total of 14.3 % of patients needed repeat endoscopic drainage and another 14.3% required surgical drainage of the pancreatic duct. Krafft et al. described a dual-center study of 28 patients undergoing anterograde EUS- PGS for chronic pancreatitis or PJ stenosis after Whipple surgery [11]. The technical and clinical success rates were 82% (23/28) and 77% (17/22), respectively, with an adverse event rate of 14 % (4/28).

### 5.1. Long-Term Outcomes

Fujii et al. [12] published long-term follow-up of 29 patients who successfully underwent EUS- PDD. Of 23 patients who successfully underwent EUS- PDD and had > 1-year follow-up, 16 patients (69.6%) had complete symptom resolution. Stents were removed after a median of 4 months in 23 patients, and symptom recurrence was seen in 13.1 % (n = 4) of patients after a median of 14 months follow-up (range 2–45 months). No further surgical or endoscopic intervention was required for these patients who had symptom recurrence or incomplete symptom resolution. Tellez et al. reported long-term results of permanent indwelling transmural stents for patients with disconnected pancreatic duct syndrome [43]. Technical success was 100% with clinical success of 80.9% (17/21 patients). A total of 25.3% of the cohort developed adverse events, most of which were stent migration.

An international multi-center prospective study reported the outcomes of 80 patients who underwent EUS- PDD [5]. A total of 83% (n = 66) of this cohort had malignant disease and 45% (n = 36) had surgically altered anatomy. Technical success was achieved in 89% (n = 71) and clinical success in 81% of patients (n = 65). The immediate adverse event rate was 20% (n = 16). A total of 12 out of 16 immediate adverse events were classified as major and included post-ERCP pancreatitis, pancreatic fluid collection, pancreatic duct leakage, and bowel perforation. The method of approach (either antegrade or retrograde) did not predict technical success or clinical success [5]. Uchida et al. compared outcomes of EUS- PGS for patients with benign strictures compared to malignant obstructions. Technical success (75% vs. 100%) and clinical success (100% vs. 85.7%) were similar for EUS- PGS performed for benign and malignant indications respectively with a non-significant difference in adverse event rates [25]. 

Recently, Sakai et al. described the endoscopic outcomes following endoscopic transpapillary pancreatic drainage (ETPD) and EUS- PDD in patients with benign pancreatic duct obstruction [45]. Eight out of ten patients who failed ETPD underwent EUS- PDD together with two patients undergoing EUS- PDD as a primary procedure. When added to ETPD, EUS- PDD improved the technical success rates of endoscopic intervention from 82% to 91% for chronic pancreatitis and 0 % to 80% in patients with pancreatico-jejunostomy stricture [45]. The overall clinical success rate of endoscopic interventions was 97%. Post-procedural pancreatitis in the EUS- PDD group was 30% (n = 3). Sakai’s study demonstrated that EUS- PDD was more likely to achieve technical success than ETPD for PJ stenosis; when added to ETPD as a salvage procedure, decreased the number of patients who would otherwise require surgical drainage for pancreatic obstruction [45].

### 5.2. Post-Operative Pancreatic Fistula

In a study of 24 patients undergoing POPF drainage after pancreatic resection, the POPF could be visualized on EUS from the gastric position in five patients and hence underwent EUS- transmural drainage (TMD). The remaining 19 patients underwent percutaneous drainage. Both EUS- TMD and percutaneous drainage achieved a technical success of 100%. The short- and long-term clinical success rates of EUS- TMD were both 100%, compared to 61.1% and 83% for percutaneous drainage. The time until clinical success for EUS- TD was markedly shorter (5.8 days vs. 30.4 days, *p* = 0.0013) in patients undergoing EUS- TMD [46].

### 5.3. Comparison between e-ERP vs. EUS- PDD 

Chen et al. compared the efficacy of EUS- PDD compared to enteroscopy-assisted ERP (e-ERP) in an international multi-center comparative retrospective study. A total of 75 procedures (40 EUS- PDD and 35 e-ERP) were performed in 66 patients. Technical (92.5% vs. 20%, *p* < 0.001) and clinical success (87.5% vs. 23.1%, *p* < 0.001) were significantly superior in the EUS- PDD group compared to e-ERP but resulted in more adverse events (35.0% vs. 2.9%, OR 18.3, *p* < 0.01) [42]. The lower technical success in the e-ERP group was due to failed cannulation in 42.9%, failed identification of PJ anastomosis (35.7%), and inability to reach the PJ in 21.4% [42]. Kogure et al. reported the outcomes of pancreatic interventions performed via double-balloon enteroscopy (DBE- assisted) ERP compared to EUS- PDD [47]. EUS- PDD was utilized as a salvage procedure when DBE -ERP failed and vice versa. The technical success of DB-ERP was 70.7% compared to 100% for EUS- PDD (*p* = 0.092). The clinical success of DB- ERP was similar to EUS- PDD (68.3% vs. 66.7%) and overall clinical success improved to 85.0% by combining both DB-ERP and EUS- PD [47].

### 5.4. Comparison between EUS- RV and EUS- TMD

Dalal et al. compared the outcomes of patients undergoing EUS-guided rendezvous technique compared to those who underwent antegrade stenting. Technical success for EUS- RV was 95.6% (22/23 patients) compared to 77.8% (14/18) in EUS- PGS (*p* = 0.08). Clinical success was also similar between the two techniques (RV 86.9% vs. PGS 72.2%). The rendezvous technique had a non-significant reduction in adverse events compared to EUS-guided PGS (17.4% vs. 33.3 %, *p* > 0.05) [16].

### 5.5. Overall Outcomes from Meta-Analyses and Systemic Reviews

Published reports currently estimate the technical success of EUS- PDD to be around 80% with an adverse event rate of 20% [14,18,48]. Imoto et al. summarized in a review the outcomes of 401 patients who underwent EUS-guided transmural stenting. The overall technical and clinical success rates were 85% (339/401 patients) and 88% (328/372 patients) respectively [19]. Adverse outcomes occurred in 25% (102/401 patients), of which 5% (20/401) were classified as severe adverse events. Bhurwal et al. summarized the outcomes of EUS-guided pancreatic duct decompression in a recent meta-analysis [48]. In this meta-analysis comprising 503 patients, the technical success rate was 81.4%, clinical success was 84.6% with an overall adverse event rate of 21.3% (mostly post-procedural pain), and pooled event rate of 5% for EUS- PD pancreatitis. Results from an earlier meta-analysis [14] were similar with a pooled technical success rate of 84.8% (95% CI 79.1–89.2) and clinical success of 89.2% (95% CI 82.1–93.7). Pooled adverse event rates were 18.1% (95% CI 14.2–22.9) and 6.6% (95% CI 4.5–9.4) for acute pancreatitis, 4.1% (95% CI 2.7–6.2) for bleeding, 3.1% (95% CI 1.9–5) for perforation and 2.3% (95% CI 1.4–4) for pancreatic leakage.

In a systemic review comparing pancreatic duct cannulation outcomes of patients who underwent ERP guided vs. EUS-guided pancreatic access for pancreatico-jejunostomy (PJ) stenosis, an EUS-guided approach resulted in higher pancreatic duct opacification (87% vs. 30%, *p* <0.001), cannulation success (79% vs. 26%, *p* < 0.001), and stent placement (72% vs. 20%, *p*< 0.001) [49]. Clinical success was also higher in the EUS group compared to the ERP group (79% vs. 19%, *p* < 0.001) [49] even though the definition of clinical success was not standardized. 

## 6. EUS- PDD Training

Tyberg et al. reported the learning curve of EUS- PDD for a single expert ERCP / EUS operator from a retrospective registry [50]. In a series comprising 56 patients, the median procedural time was found to be 80 min (range 49–159 min). CUSUM analysis showed a progressive reduction in procedural time with a procedural time of 80 min achieved on the 27th procedure, indicating procedural efficiency. Procedural duration further reduced until the 40th procedure before reaching a plateau indicating proficiency. These results suggest that even for the experienced interventional endoscopist, 40 cases are required prior to mastery of the procedure [50]. Technical success was achieved in 84% of patients and the overall adverse event rate was 24% in this series [50].

## 7. Future Directions

Over the last two decades, EUS- PDD has evolved significantly and now plays an important role in the management of pancreatic duct obstruction, especially in patients with altered surgical anatomy and failed drainage by traditional ERP techniques. Case series regarding EUS- PDD have mostly been retrospective in nature involving small cohorts of patients. Consequently, guidelines surrounding EUS- PDD have only provided guiding statements backed up by low-quality evidence [51,52]. There is a need for larger prospective and long-term comparative studies in the field of EUS- PDD. Several studies consist of mixed cohorts of EUS- PDD patients who have undergone EUS- RV and transmural stenting and outcomes are not reported separately [4,5,36]; future studies should report the outcomes of EUS- RV and transmural stenting independently as these procedures have varying technical considerations as well as varying technical success. Furthermore, definitions for technical and clinical success in EUS- PDD have not been standardized. Technical success in EUS- PDD varies and has been defined as a successful pancreatogram, successful negotiation of the guidewire past the obstruction, as well as stent placement into the MPD in different studies. The lack of standardization makes the comparison of results difficult, and this is evident in the results of a meta-analysis in EUS- PDD where heterogeneity has been noted [14,48]. EUS- PDD is a challenging procedure due to several technical factors as previously discussed, as well as the lack of dedicated accessories and limited training opportunities due to the rare indications for the procedure. The development of dedicated accessories such as small caliber accessories [34] may improve the technical success of EUS- PDD but direct head-to-head comparisons with standard accessories have not been performed. Adverse events still occur in about 20% of EUS- PDD cases and guidelines have suggested that experience in other EUS-guided drainage procedures may improve the success rates and reduce the risk of adverse events [52]. Given that the learning curve of EUS- PDD has recently been described to be around 40 cases [50], it is imperative to research how the EUS- PDD learning curve can be surmounted in fewer cases through better standardized training and the provision of better accessories to improve the success rates and safety of the procedure. 

The advent of per-oral pancreatoscopy is an exciting development in EUS-guided pancreatic duct access. Currently, there are few case series proving the efficacy and safety of POPS [35]; per oral cholecystoscopy after EUS- gallbladder drainage opened a whole new paradigm into the treatment of gallbladder-related diseases, such as cholecystoscopy-guided target biopsy of gallbladder neoplasm and lithotripsy of gallbladder stones [53]. Similarly, POPS can expand the indications for EUS-guided PD access and allow the development of a new tool for luminal diagnosis and management of pancreatic diseases.

## 8. Conclusions

EUS- PDD is a valuable skill in the interventional endoscopist’s armamentarium of skills to deal with main pancreatic duct obstruction when traditional ERP techniques have failed. It is especially valuable in cases with altered surgical anatomy, such as in PJAS after the Whipple operation. In skilled hands, EUS- PDD is associated with high technical and clinical success rates, although one in five patients who undergo EUS- PDD may still experience adverse events. Standardization of EUS- PDD techniques, the introduction of dedicated accessories, and the provision of structured training in high-volume centers will improve the outcomes of patients who undergo EUS- PDD who might otherwise require surgical management which is associated with high morbidity and mortality.

## Figures and Tables

**Figure 1 jcm-12-01626-f001:**
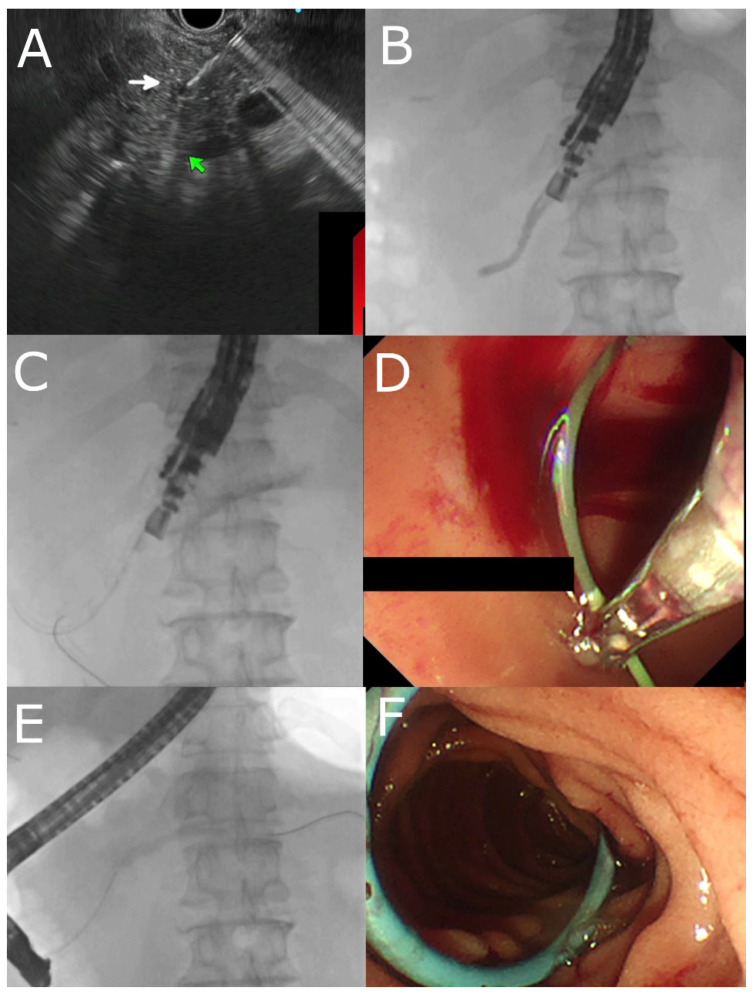
EUS-guided rendezvous ERP in a patient with pancreatic divisum and failed minor papilla cannulation. (**A**) EUS-guided puncture of the pancreatic duct (indicated by white arrow). (**B**) Contrast injection showing pancreatogram. (**C**) Passage of guidewire into the duodenum. (**D**) Grabbing the guidewire from the duodenal lumen after changing to a duodenoscope with micro-forceps. (**E**) Main pancreatic duct recannulated with a guidewire. (**F**) Insertion of a pancreatic duct stent after minor papillotomy.

**Figure 2 jcm-12-01626-f002:**
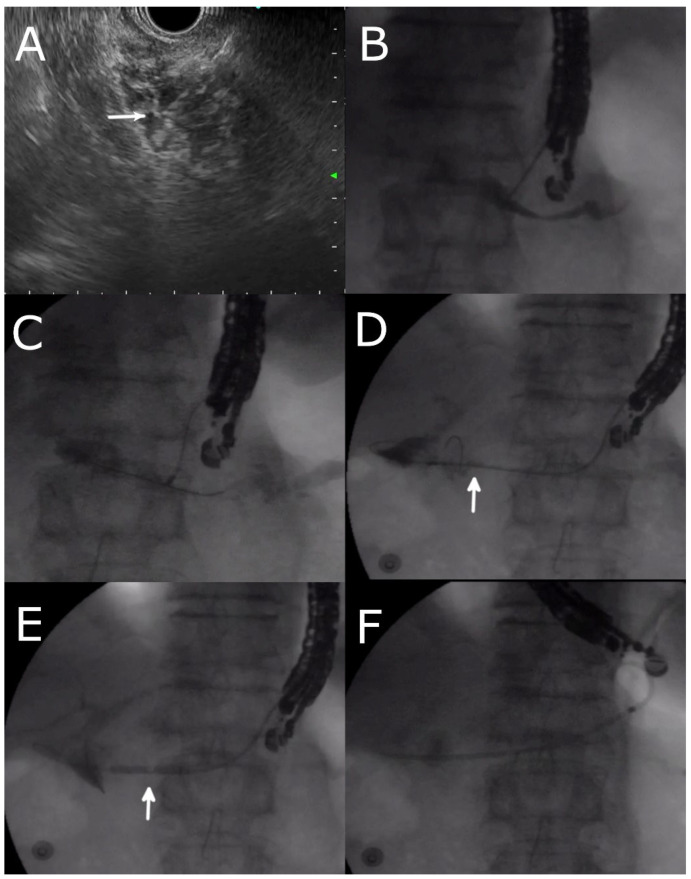
EUS-guided drainage of a pancreatico-jejunal anastomotic stricture after Whipple operation. (**A**) EUS-guided puncture of the pancreatic duct (arrow indicates the position of the needle) (**B**) Injection of contrast through the needle showing a pancreatogram. (**C**) Passage of guidewire and cystotome into the pancreatic duct. (**D**) Passage of the guidewire across the anastomotic stricture (indicated by the arrow) with the use of the cystotome as a pivot. Contrast injection through the cystotome outlined the jejunum and also part of the bile duct. (**E**) Dilation of the stricture with a 4 mm balloon (indicated by the arrow). (**F**) Placement of a plastic stent across the PJ anastomosis into the stomach.

**Table 1 jcm-12-01626-t001:** Indications and contraindications of EUS- PDD drainage.

*Indications*
**Native Anatomy (usually after failed ERP)**
Chronic pancreatitis and main pancreatic duct (MPD) obstruction
Symptomatic pancreatic stones
Disconnected Pancreatic Duct Syndrome
**Surgically Altered Anatomy/Inaccessible PD**
Pancreatico-jejunostomy anastomosis stricture post Whipple operation
Standard ERP indications with history of previous billroth II/Roux-en-Y Gastrectomy
Inaccessible papilla due to malignant/benign duodenal strictures
*Contraindications*
Technical Factors
Inability to locate the MPD on EUS
Insufficient dilatation of MPD (MPD size < 4 mm)
Intervening vessels at the puncture site
Long distance between bowel and pancreatic duct
Multi-level strictures
Patient factors
Hemodynamic instability
Uncorrected coagulopathy, thrombocytopenia

**Table 2 jcm-12-01626-t002:** Outcomes of EUS-guided pancreatic duct intervention.

Author	Type of Study	Patients	Indications	Technical Success, n (%)	Clinical Success, n (%)	Adverse Events, n (%)	Comments
Kahaleh et al. (2007) [4]	Prospective	13 [EUS- TMD]	SAA, strictures secondary to pancreatitis, IPMN	10/13 (77%)	NR	2/13 (15.3%) Bleeding (1), perforation (1)	Improvement in MPD diameter, pain score, and weight on long term follow up
Tessier et al. (2007) [37]	Retrospective	36 [EUS- TMD]	SAA, chronic pancreatitis, PJAS	33/36 (92%)	Pain relief: 25/36 (69%) Stent dysfunction: 20/36 (55%)	5/35 (13.8%) 2 severe, 3 mild	
Barkay et al. (2010) [38]	Retrospective	21 [EUS- RV]	Failed ERP	10/21 (48%)	NR	2/20 (10%) 1 case of pancreatitis 1 case of peripancreatic abscess)	Dilated PD was associated with greater likelihood of EUS-guided pancreatography
Ergun et al. (2011) [39]	Retrospective	20 [total]/ 24 procedures 5 [EUS- RV] 19 [EUS- TMD]	CP, PJAS	18/20 (90%) 5/5 (100%) 15/19 (79%)	Pain term pain resolution: 13/18 (72%)	2/20 (10%) including bleeding and perigatric collection. 9/18 (50%) developed stent dysfunction	Significant decrease in pain scores and MPD size after long-term follow-up.
Shah et al. (2012) [40]	Retrospective	24 [total]/30 procedures 16 [EUS- RV] 14 [EUS- TMD]	CP, pancreatic duct leak, PJAS	19/30 (63%) 9/16 (56%) 10/14 (71%)	NR	4/ 22 (18%)	
Kurihara et al. (2013) [20]	Retrospective	14 [total]/17 procedures 11 [EUS- RV] 5 [EUS- TMD]	PJAS, CP	14/17 (82.3%) 11/17 (64.7%) 3/5 (60%)	NR	1/17 (5.8%) 1 case developed pancreatic pseudocyst with aneurysm	Patients underwent EUS- PD after failed EUS- RV.
Fujii et al. (2013) [12]	Retrospective	45 [total]	SAA, failed ERP	32/43 (74%) 14 [EUS- RV] 18 [EUS- TMD]	Long-term symptom resolution: 24/29 (83%)	3/45 (6.6%) with severe complications 16/35 (35.5%) developed abdominal pain	EUS- RV significantly longer than EUS- TMD (130 vs. 125 min, *p* = 0.05)
Will et al. (2015) [41]	Retrospective	94 [total]/111 procedures	CP, pancreatic divisum, DPDS, POPF	47/83 (56.6%) 21 [EUS- RV] 26 [EUS- TMD]	68/83 (81.9%)	24/111 (21.6%) (2 severe, 20 intermediate, 2 minor AEs)	
Chen et al. (2017) [42]	Retrospective	40 [Total] 37 [EUS- TMD] 3 [EUS- RV]	Pancreatic intervention post-Whipple operation	37/40 (92.5%) 34/ 37 (91.8%) 3/3 (100%)	32/40 (87.5%) 29/37 (78.3%) 3/3 (100%)	14/40 (35.0%)	
Tyberg et al. (2018) [5]	Retrospective	80 [total] 66 [EUS- RV] 14 [EUS- TMD]	Malignancy, chronic pancreatitis	71/ 80 (89%)	65/80 (81%)	Immediate 16/80 (20%); Delayed 9/80 (11%)	Comparative study of EUS- PDD and e-ERP. EUS- PDD had higher clinical and technical success.
Uchida et al. (2018) [25]	Retrospective	15 [total] 2 [EUS- RV] 13 [EUS- TMD]	Pancreatic strictures (8 benign, 7 malignant)	13/15 (86%) Benign 75% (6/8) malignant 100% (7/7)	12/13 (92.3%) Benign 100% (6/6), malignant 87.5% (6/7)	4/15 (26.7%)–peritonitis, stent migration, bleeding	
Tellez-Avina et al. (2018) [43]	Retrospective	21 [EUS- TMD]	DPDS	21/21 (100%)	17/21 (80.9%)	5/21 (23.8%)	
Matsunami et al. (2019) [44]	Retrospective	30 [EUS- TMD]	Acute recurrent pancreatitis with stricture	30/30 (100%)	23/30 (76%)	7/30 (23%): mild abdominal pain/bleeding/pancreatitis 6/25 (24%): stent dislodgement	
Oh et al. (2019) [33]	Retrospective	23 [total] 3 patients underwent plastic stenting 20 patients underwent FCSEMS	PJAS	23/23 (100%)	23/23 (100%)	Early adverse events: 4/23 (17.4%) Late adverse events: 5/23 (21.7%)	Utilized FCSEMS.
Krafft et al. (2020) [11]	Retrospective	28 [EUS- TMD]	CP, PJS	23/28 (82%)	21/28(75%)	4/28 (14.2%)	Long-term outcomes: 52% developed DM, 14.2% developed exocrine insufficiency, 83% had stents in situ after 12 months
Dalal et al. (2020) [16]	Retrospective	44 [total] 23/44 [EUS- RV] 21/44 [EUS- TMD]	Failed ERP, SAA	39/44 (88.6%) 22/23 (95.6%) 17/21 (80.9%)	35/44 (79.5%) 19/23 (82.6%) 16/21 (76.1%)	10/44 (22.7%)	2/28 patients underwent gastropancreaticoenterostomy “ring drainage”

Legend: SAA–surgically altered anatomy, IPMN–intraductal papillary mucinous neoplasm, MPD–main pancreatic duct, PJAS–pancreatico-jejunal anastomotic strictures, ERP–endoscopic retrograde pancreatography, CP–chronic pancreatitis, DPDS–disconnected pancreatic duct syndrome, POPF–post-operative pancreatic fistula, EUS- RV–EUS rendezvous, EUS- TMD–EUS-transmural stenting, FCSEMs–fully covered self-expanding metal stents.

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
