# Peer review of "Techniques and Outcomes of Endoscopic Ultrasound Guided—Pancreatic Duct Drainage (EUS- PDD)"

_jcm, 2023, doi:10.3390/jcm12041626_

Round 1
Reviewer 1 Report
The authors present a narrative review on Endoscopic ultrasound guided-pancreatic duct drainage. The paper is well written, and the topic is interesting. Considering the low frequency of this type of procedures, limited to few referral centers, the paper should be considered for publication in gastroenterology and/or endoscopy specific journals.
Indeed, there are several major issues that need to be addressed.
1. In the “Indications for EUS-PDD” chapter, the authors report that fibrosis and inflammation in acute recurrent pancreatitis leading to MPD obstruction might be included as indication for EUS-PDD. This reviewer thinks that MPD obstructions due to inflammation should be classified as chronic pancreatitis. Reporting recurrent pancreatitis as indication to the procedure might be confusing for readers. Please modify/delete this part.
2. In the same chapter, please specify that the need of MPD drainage in malignant MPD stricture is extremely rare, considering that symptoms are related to the malignant disease and infiltration more than to MPD obstruction.
3. In Table 1:
- please deleted acute recurrent pancreatitis and malignant causes from indications;
- in contraindication please clarify what is insufficient MD dilation (< 4-5 mm); please comment on this in the text;
- in “contraindications” please delete “no previous ERP attempts” considering that in specific post-surgical scenarios EUS-PDD might be considered as first approach (Garcia-Alonso et al.).
4. In the Chapter “patients preparation” the authors report how they perform EUS-PDD, including the need for prophylactic antibiotics, etc. In the review the authors should include information based on published data and not their own experience. Therefore, the chapter and the single information included need to have specific references. Otherwise, this part should be delated.
5. In the Chapter “Approaches and equipment” the authors report that EUS-RV is the procedure of choice if the papilla is accessible. Is this the opinion of the authors? Are there some data suggesting this approach? Please report references and data supporting this. Moreover, in the same chapter, authors report that “stenting performed by EUS-TMD can be either via EUS-guided pancreatico-enterostomy or trans-papillary” Why only pancreatico- enterostomy and not pancreatico-gastrscopy?
6. In the chapter “Pancreatic duct access” the author specified the brand of echoendoscope and accessories. This reviewer thinks that there is no need for this, and no brand should be reported but just features sizes and general characteristics of the devices. The same for the following chapters of the paper.
7. In the “Transmural approaches with transpapillary or trans-anastomotic stenting” chapter, authors report again “EUS-guided pancreatico-enterostomy” Why only pancreatico- enterostomy and not pancreatico-gastrscopy?
8. In Table 2 please report adverse event as number and percentage for every single study included.
Minor issues:
1. In “Long ther outcomes2 chpater, at line 277 deleted the repetition “in patients in patients”
Author Response
Response to reviewers comments
Reviewer 1
The authors present a narrative review on Endoscopic ultrasound guided-pancreatic duct drainage. The paper is well written, and the topic is interesting. Considering the low frequency of this type of procedures, limited to few referral centers, the paper should be considered for publication in gastroenterology and/or endoscopy specific journals.
Indeed, there are several major issues that need to be addressed.
- In the “Indications for EUS-PDD” chapter, the authors report that fibrosis and inflammation in acute recurrent pancreatitis leading to MPD obstruction might be included as indication for EUS-PDD. This reviewer thinks that MPD obstructions due to inflammation should be classified as chronic pancreatitis. Reporting recurrent pancreatitis as indication to the procedure might be confusing for readers. Please modify/delete this part.
Answer: Recurrent acute pancreatitis as an indication for EUS – PDD has been deleted in the text as suggested by reviewer 1
- In the same chapter, please specify that the need of MPD drainage in malignant MPD stricture is extremely rare, considering that symptoms are related to the malignant disease and infiltration more than to MPD obstruction.
Answer: We have added on “Pancreatic outflow obstruction resulting in ductal hypertension are seen commonly in benign pancreatic obstruction, and less commonly in infiltrative malignant disease, where the need for pancreatic duct decompression is rare.” (lines 41- 45)
- In Table 1:
- please deleted acute recurrent pancreatitis and malignant causes from indications;
Answer: Thank you, this has been deleted from Table 1. A
- in contraindication please clarify what is insufficient MD dilation (< 4-5 mm); please comment on this in the text;
Answer: Thank you to Reviewer 1 for his suggestion. We have included <4mm as contraindication for EUS – PDD in table 1. Furthermore, we have expanded on the discussion on degree of MPD dilatation for successful PD puncture between lines 73- 85.
- in “contraindications” please delete “no previous ERP attempts” considering that in specific post-surgical scenarios EUS-PDD might be considered as first approach (Garcia-Alonso et al.).
Answer: This has been deleted as suggested, thank you.
- In the Chapter “patients preparation” the authors report how they perform EUS-PDD, including the need for prophylactic antibiotics, etc. In the review the authors should include information based on published data and not their own experience. Therefore, the chapter and the single information included need to have specific references. Otherwise, this part should be delated.
Answer: References to the section have been added as per the reviewer’s comments.
- In the Chapter “Approaches and equipment” the authors report that EUS-RV is the procedure of choice if the papilla is accessible. Is this the opinion of the authors? Are there some data suggesting this approach? Please report references and data supporting this. Moreover, in the same chapter, authors report that “stenting performed by EUS-TMD can be either via EUS-guided pancreatico-enterostomy or trans-papillary” Why only pancreatico- enterostomy and not pancreatico-gastrscopy?
Answer: Thank you for the comments. EUS – RV is the procedure of choice for PD drainage in native anatomy due to high success rates, less complications and avoidance of extra- anatomical stent. This was been added into the text and reference added. We have further clarified that EUS – TMD can be performed through a transduodenal or transgastric approach and the nomenclature for these procedures have also been introduced in this paragraph and standardized in the text.
- In the chapter “Pancreatic duct access” the author specified the brand of echoendoscope and accessories. This reviewer thinks that there is no need for this, and no brand should be reported but just features sizes and general characteristics of the devices. The same for the following chapters of the paper.
Answer: We have removed the brand of the echoendoscope but kept the brand names of the different accessories as readers may be interested in the array of accessories used to perform EUS -PDD .
- In the “Transmural approaches with transpapillary or trans-anastomotic stenting” chapter, authors report again “EUS-guided pancreatico-enterostomy” Why only pancreatico- enterostomy and not pancreatico-gastrscopy?
Answer: Thank you reviewer 1. EUS guided pancreatico-enterostomy has been changed to EUS – PDS and EUS – PGS to clarify that both approaches are used.
- In Table 2 please report adverse event as number and percentage for every single study included.
Answer: Table 2 has been updated to the relevant numbers and percentages as suggested by review 1.
Minor issues:
- In “Long ther outcomes2 chpater, at line 277 deleted the repetition “in patients in patients”
Answer: This has been corrected, thank you.
Reviewer 2
I have read with great interest the present review by Teh and Teoh. It shows the most recent evidences about indications and outcomes of the novel endoscopic procedure EUS-pancreatic duct drainage (EUS-PDD) . It also shows the steps of all the available EUS-PDD procedures (rendez-vous, transmural with transpapillary/transanastomotic stenting and transmural with antegrade/retrograde stenting)..
The paper is well written and I have only some minor comment in order to improve the fluency of reading:
- In general the nomenclature of all the available EUS-PDD procedures (rendezvous, transmural stenting) should be homogenized within the text (for example in lines 318-321 it is mentioned the antegrade stenting and then the transgastric pancreaticogastrostomy) and better explained and probably simplified in a table.
Answer: The nomenclature of EUS – PDD procedures were summarised under the chapter “approaches and equipment”. The acronyms in the text are EUS – PDD (pancreatic duct drainage), EUS – RV (Rendezvous), EUS – TMD (transmural drainage) which includes EUS – PGS (pancreaticogastrostomy) and EUS – PDS ( pancreaticoduodenostomy). The use of transgastric / transduodenal approaches in the texts are changed to EUS – PGS and EUS – PDS.
- the paragraph indication for EUS-PDD should be divided in “indication” and “controindications and limitations”. In this section i think that should be emphasized if there are data in literature about the minimum Wirsung dilation caliber in order to proceed to EUS-PDD.
Answer: We have expanded the discussion on the pre requisite for successful PD puncture under the chapter of contraindications.
- The paragraph “Long term outcome” i think should be moved at the end of the Outcome section.
Answer: We arranged the discussion of EUS – PDD outcomes to first discuss the general and long term outcomes of EUS – PDD as these cohorts consisted of patients who underwent different approaches of EUS – PDD. Subsequently, we discussed outcomes in selected scenarios and data comparing different EUS -PDD approaches. Finally we reviewed the data from meta-analyses and systemic reviews.
- The paragraph PJ anastomosis stenosis is mainly focused on the comparison between EUS-PDD and enteroscopy assisted ERP therefore i think that the paragraph should be renamed “EUS-PDD vs e-ERP” . The last lines from 309 to 315 (“In a systematic review…”) should be moved in the section of “overall outcomes from meta-analysis and reviews”.
Answer: The paragraph has been renamed as per reviewer 2’s suggestion and the last lines in that paragraph has been shifted to the meta-analysis section. Thank you.
- Move the paragraph “outcomes of POPS” in the section “Per-oral pancreatoscopy”.
Answer: This paragraph has been shifted to the section of Per-oral pancreatoscopy.
- Remove the paragraph outcomes of EUS-antegrade stenting : the paper of Imoto et al can be cited in the section of meta-analysis and reviews whereas the paper of Krafft can be cited at the beginning of the Outcome section
Answer: Thank you to reviewer 2 for the suggestion. The papers have been shifted to their relevant sections.
- Insert a figure (schematic or real EUS/radiological figures) in order to summarize the steps of transmural stenting with antegrade/retrograde stenting
Answer: Thank you. Figure 2 is the figure summarizing the steps of transmural stenting.
- Table 1: split the table 1 in indications and contraindications; the list of indications are redundant and should be better explained: the main indications are for benign or malignant situations that may occur in both normal or altered anatomy.
Answer: We separated indications into native anatomy and surgically altered anatomy as that would affect choice of EUS -PDD approach. Where native anatomy is present, EUS – RV can be attempted as the first choice procedure. We have removed malignant causes from table 1 as per reviewer 1’s suggestion.
- Table 2: it is used the acronym PDS (pancreatic duct stenting) that is never mentioned within the paper. i think it is better to use the same terminology mentioned in the paper such as EUS-TMD
Answer: This has been corrected, thank you.
- In general there are too many acronyms used within the text, especially those used for recent procedures that are not yet familiar for all the endoscopists. I think they should be reduced in order to facilitate the reader.
Answer: The nomenclature of EUS – PDD procedures were summarised under the chapter “approaches and equipment”. The acronyms in the text are EUS – PDD (pancreatic duct drainage), EUS – RV (Rendezvous), EUS – TMD (transmural drainage) which includes EUS – PGS (pancreaticogastrostomy) and EUS – PDS ( pancreaticoduodenostomy). The use of transgastric / transduodenal approaches in the texts are changed to EUS – PGS and EUS – PDS.

Reviewer 2 Report
Dear editor,
I have read with great interest the present review by Teh and Teoh. It shows the most recent evidences about indications and outcomes of the novel endoscopic procedure EUS-pancreatic duct drainage (EUS-PDD) . It also shows the steps of all the available EUS-PDD procedures (rendez-vous, transmural with transpapillary/transanastomotic stenting and transmural with antegrade/retrograde stenting)..
The paper is well written and I have only some minor comment in order to improve the fluency of reading:
- In general the nomenclature of all the available EUS-PDD procedures (rendezvous, transmural stenting) should be homogenized within the text (for example in lines 318-321 it is mentioned the antegrade stenting and then the transgastric pancreaticogastrostomy) and better explained and probably simplified in a table.
- the paragraph indication for EUS-PDD should be divided in “indication” and “controindications and limitations”. In this section i think that should be emphasized if there are data in literature about the minimum Wirsung dilation caliber in order to proceed to EUS-PDD.
- The paragraph “Long term outcome” i think should be moved at the end of the Outcome section.
- The paragraph PJ anastomosis stenosis is mainly focused on the comparison between EUS-PDD and enteroscopy assisted ERP therefore i think that the paragraph should be renamed “EUS-PDD vs e-ERP” . The last lines from 309 to 315 (“In a systematic review…”) should be moved in the section of “overall outcomes from meta-analysis and reviews”.
- Move the paragraph “outcomes of POPS” in the section “Per-oral pancreatoscopy”.
- Remove the paragraph outcomes of EUS-antegrade stenting : the paper of Imoto et al can be cited in the section of meta-analysis and reviews whereas the paper of Krafft can be cited at the beginning of the Outcome section
- Insert a figure (schematic or real EUS/radiological figures) in order to summarize the steps of transmural stenting with antegrade/retrograde stenting
- Table 1: split the table 1 in indications and contraindications; the list of indications are redundant and should be better explained: the main indications are for benign or malignant situations that may occur in both normal or altered anatomy.
- Table 2: it is used the acronym PDS (pancreatic duct stenting) that is never mentioned within the paper. i think it is better to use the same terminology mentioned in the paper such as EUS-TMD
- In general there are too many acronyms used within the text, especially those used for recent procedures that are not yet familiar for all the endoscopists. I think they should be reduced in order to facilitate the reader.
Author Response
Reviewer 2
I have read with great interest the present review by Teh and Teoh. It shows the most recent evidences about indications and outcomes of the novel endoscopic procedure EUS-pancreatic duct drainage (EUS-PDD) . It also shows the steps of all the available EUS-PDD procedures (rendez-vous, transmural with transpapillary/transanastomotic stenting and transmural with antegrade/retrograde stenting)..
The paper is well written and I have only some minor comment in order to improve the fluency of reading:
- In general the nomenclature of all the available EUS-PDD procedures (rendezvous, transmural stenting) should be homogenized within the text (for example in lines 318-321 it is mentioned the antegrade stenting and then the transgastric pancreaticogastrostomy) and better explained and probably simplified in a table.
Answer: The nomenclature of EUS – PDD procedures were summarised under the chapter “approaches and equipment”. The acronyms in the text are EUS – PDD (pancreatic duct drainage), EUS – RV (Rendezvous), EUS – TMD (transmural drainage) which includes EUS – PGS (pancreaticogastrostomy) and EUS – PDS ( pancreaticoduodenostomy). The use of transgastric / transduodenal approaches in the texts are changed to EUS – PGS and EUS – PDS.
- the paragraph indication for EUS-PDD should be divided in “indication” and “controindications and limitations”. In this section i think that should be emphasized if there are data in literature about the minimum Wirsung dilation caliber in order to proceed to EUS-PDD.
Answer: We have expanded the discussion on the pre requisite for successful PD puncture under the chapter of contraindications.
- The paragraph “Long term outcome” i think should be moved at the end of the Outcome section.
Answer: We arranged the discussion of EUS – PDD outcomes to first discuss the general and long term outcomes of EUS – PDD as these cohorts consisted of patients who underwent different approaches of EUS – PDD. Subsequently, we discussed outcomes in selected scenarios and data comparing different EUS -PDD approaches. Finally we reviewed the data from meta-analyses and systemic reviews.
- The paragraph PJ anastomosis stenosis is mainly focused on the comparison between EUS-PDD and enteroscopy assisted ERP therefore i think that the paragraph should be renamed “EUS-PDD vs e-ERP” . The last lines from 309 to 315 (“In a systematic review…”) should be moved in the section of “overall outcomes from meta-analysis and reviews”.
Answer: The paragraph has been renamed as per reviewer 2’s suggestion and the last lines in that paragraph has been shifted to the meta-analysis section. Thank you.
- Move the paragraph “outcomes of POPS” in the section “Per-oral pancreatoscopy”.
Answer: This paragraph has been shifted to the section of Per-oral pancreatoscopy.
- Remove the paragraph outcomes of EUS-antegrade stenting : the paper of Imoto et al can be cited in the section of meta-analysis and reviews whereas the paper of Krafft can be cited at the beginning of the Outcome section
Answer: Thank you to reviewer 2 for the suggestion. The papers have been shifted to their relevant sections.
- Insert a figure (schematic or real EUS/radiological figures) in order to summarize the steps of transmural stenting with antegrade/retrograde stenting
Answer: Thank you. Figure 2 is the figure summarizing the steps of transmural stenting.
- Table 1: split the table 1 in indications and contraindications; the list of indications are redundant and should be better explained: the main indications are for benign or malignant situations that may occur in both normal or altered anatomy.
Answer: We separated indications into native anatomy and surgically altered anatomy as that would affect choice of EUS -PDD approach. Where native anatomy is present, EUS – RV can be attempted as the first choice procedure. We have removed malignant causes from table 1 as per reviewer 1’s suggestion.
- Table 2: it is used the acronym PDS (pancreatic duct stenting) that is never mentioned within the paper. i think it is better to use the same terminology mentioned in the paper such as EUS-TMD
Answer: This has been corrected, thank you.
- In general there are too many acronyms used within the text, especially those used for recent procedures that are not yet familiar for all the endoscopists. I think they should be reduced in order to facilitate the reader.
Answer: The nomenclature of EUS – PDD procedures were summarised under the chapter “approaches and equipment”. The acronyms in the text are EUS – PDD (pancreatic duct drainage), EUS – RV (Rendezvous), EUS – TMD (transmural drainage) which includes EUS – PGS (pancreaticogastrostomy) and EUS – PDS ( pancreaticoduodenostomy). The use of transgastric / transduodenal approaches in the texts are changed to EUS – PGS and EUS – PDS.

Round 2
Reviewer 1 Report
This Reviewer thinks that the issues have been addressed.